# Comparison of the Performance of the PanBio COVID-19 Antigen Test in SARS-CoV-2 B.1.1.7 (Alpha) Variants versus non-B.1.1.7 Variants

M. L. van Ogtrop,ᵃ T. J. W. van de Laar,ᵃ,ᵇ D. Eggink,ᶜ J. W. Vanhommerig,ᵈ W. A. van der Reijdenᵃ,ᵉ

ᵃDepartment of Clinical Microbiology, OLVG Lab BV, Amsterdam, The Netherlands
ᵇDepartment of Donor Medicine Research, Laboratory of Blood-borne Infections, Sanquin Research, Amsterdam, The Netherlands
ᶜCentre for Infectious Disease Control, WHO COVID-19 Reference Laboratory, National Institute for Public Health and the Environment (RIVM), Bilthoven, The Netherlands
ᵈDepartment of Research and Epidemiology, OLVG Hospital, Amsterdam, The Netherlands
ᵉDepartment of Clinical Microbiology, COMICRO, Hoorn, The Netherlands

**ABSTRACT**   This study evaluates the performance of the PanBio COVID-19 antigen (Ag) test as part of a hospital infection control policy. Hospital staff was encouraged to get tested for COVID-19 when presenting with SARS-CoV-2-related symptoms. In a period of approximately 5 months, a steady decline in the performance of the Ag test was noted, epidemiologically coinciding with the rise of the SARS-CoV-2 B.1.1.7 (alpha) variant of concern (VOC) in the Netherlands. This led to the hypothesis that the diagnostic performance of the PanBio COVID-19 Ag test was influenced by the infecting viral variant. The results show a significantly lower sensitivity of the PanBio COVID-19 Ag test in persons infected with the B.1.1.7 (alpha) variant of SARS-CoV-2 in comparison with that in persons infected with non-B.1.1.7 variants, also after adjustment for viral load.

**IMPORTANCE** Antigen tests for COVID-19 are widely used for rapid identification of COVID-19 cases, for example, for access to schools, festivals, and travel. There are several FDA- and CE-cleared tests on the market. Their performance has been evaluated mainly on the basis of infections by the classical variant of the causing virus, SARS-CoV-2. This paper provides evidence that the performance of one of the most widely used antigen tests detects significantly fewer cases of COVID-19 by the alpha variant than by the classical variants of SARS-CoV-2. This means that the role of antigen tests needs to be reevaluated in regions where other variants of SARS-CoV-2 predominate.

**KEYWORDS** SARS-CoV-2, COVID-19, antigen, diagnostics, B.1.1.7 variant

The SARS-CoV-2 pandemic has led to disastrous effects on the economy and health care system of affected countries. Aside from vaccination, the most effective preventive measure against spread of the infection is isolation of infectious individuals from susceptible individuals. For this purpose, simple, cheap, and rapid point-of-care tests are needed (1). A number of COVID-19 antigen (Ag) tests have been approved for use as rapid screening tests and could be used for effective source control. As part of a hospital infection control program, the PanBio COVID-19 Ag test was evaluated in an effort to effectively prevent the nosocomial spread of COVID-19. The PanBio COVID-19 Ag test was compared to the standard nucleic acid amplification test (NAAT) in our institute.

## RESULTS

Between 11 November 2020 to 20 April 2021, $n = 1,033$ combined SARS-CoV-2 Ag and SARS-CoV-2 RNA tests were performed on health care workers (HCWs) working at BovenIJ hospital. In total, 47 HCWs (4.6%) tested SARS-CoV-2 Ag positive and 75 HCWs

Address correspondence to M. L. van Ogtrop, m.l.vanogtrop@olvg.nl.

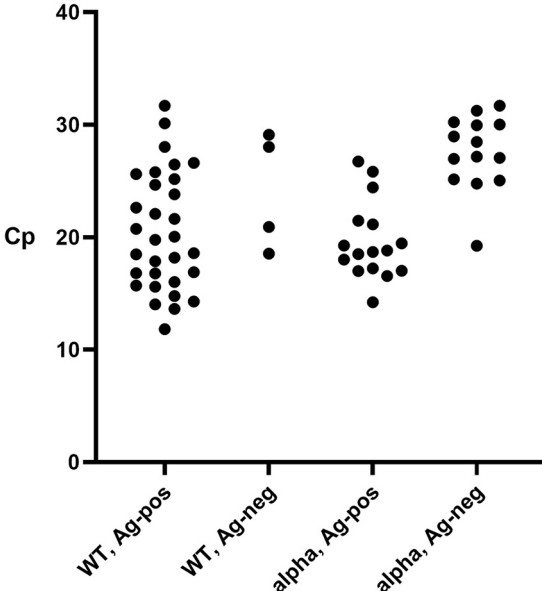

**FIG 1** Distribution of Cp values in the non-B.1.1.7 (WT) and B.1.1.7 variants (alpha) and the antigen-positive (Ag-pos) and antigen-negative (Ag-neg) samples.

(7.3%) tested SARS-CoV-2 RNA positive using transcription-mediated amplification (TMA). All 47 SARS-CoV-2 Ag-positive HCWs also tested positive for SARS-CoV-2 RNA using TMA, and the remaining 28 SARS-CoV-2 RNA-positive samples tested SARS-CoV-2 Ag negative. Real-time PCR (RT-PCR) confirmed the presence of SARS-CoV-2 RNA in all 75 TMA-positive samples. Some HCWs were tested more than once, but none of them had more than one SARS-CoV-2 RNA-positive sample. We did not include sequential samples of one viremic episode, and we did not observe episodes of reinfections. Using NAAT as the gold standard, there were no false-positive Ag tests, indicating 100% specificity of the SARS-CoV-2 Ag test. Overall sensitivity of the COVID-19 Ag test was 62.7% (47/75 SARS-CoV-2 RNA-positive HCWs were detected using Ag testing). However, the sensitivity of Ag testing decreased over time from 90.1% in November 2020, to 82.4% in December 2020, to rates varying between 66.7% and 37.5% in January to April 2021. We speculated that aside from viral load (as indicated by the manufacturer), the circulating viral variant might also influence the sensitivity of the SARS-CoV-2 Ag test. Figure 1 shows the distribution of crossing point (Cp) values for Ag-positive versus Ag-negative test results, separated per variant.

SARS-CoV-2 B.1.1.7 typing was performed on all 75 PCR-positive samples. In period 1 (November 2020 to January 2021), we obtained 48 SARS-CoV-2 RNA-positive samples. Based on the N501Y VirSNiP method, 33 SARS-CoV-2 strains lacked the N501Y mutation, 7 strains had the N501Y mutation, and 8 samples could not be typed due to (very) low viral loads (Cp value: 31.3 to 39.1). In period 2, we obtained 27 SARS-CoV-2 RNA-positive samples. Full-genome sequencing revealed B.1.1.7 ($n = 23$), B.1.351 ($n = 1$), B.1.258.21 ($n = 1$), and B.1.177.73 ($n = 1$), and one sample could not be typed (Cp value of 36.3). Hence, over the total study period, we obtained 36 confirmed non-B.1.1.7 strains (NGS plus VirSNiP), of which one was a B.1.351 strain that bore the N501Y mutation, 23 confirmed B.1.1.7 strains (NGS), 7 suspected B.1.1.7 strains (VirSNiP), and 9 non-typed low viremic samples. To enable a comparison between B.1.1.7 variants versus non-B.1.1.7 variants lacking N501Y, we excluded the 9 non-typed strains plus the N501Y-bearing B.1.351 (beta) strain, leaving a total of 65 samples for comparison.

The performance of the PanBio COVID-19 Ag test for B.1.1.7 variants versus non-B.1.1.7 variants is shown in Table 1. The sensitivity of the Ag test was 89% for non-B.1.1.7-variants and 53% for B.1.1.7 variants. The difference in performance between

**TABLE 1** Comparison of performance of the PanBio COVID-19 Ag test in infections by the B.1.1.7 variant and the non-B.1.1.7 variants

| Variant type | No. antigen-positive | No. antigen-negative |
|---|---|---|
| Non-B.1.1.7 | 31 | 4 |
| B.1.1.7 | 16 | 14 |

the two groups was statistically significant ($P = 0.002$, Fisher's exact test). The average Cp value of the non-B.1.1.7 samples was 20.9 (standard deviation [SD] 5.3), and that of the B.1.1.7 variant was 23.3 (SD 5.2). The difference was not statistically significant ($P = 0.065$, Student's $t$ test). Univariable logistic regression analysis revealed an odds ratio (OR) of 6.8 (95% confidence interval [CI]: 1.9 to 23.8; $P = 0.003$) for the chance of a B.1.1.7 variant being negative in the Ag test compared to that of the non-B.1.1.7 variant being negative. When corrected for Cp value, the adjusted OR was 7.4 (95% CI: 1.6 to 34.5; $P = 0.011$).

## DISCUSSION

Our study showed a decreasing sensitivity of the PanBio COVID-19 Ag test over time, which was traced back to the rapid emergence of the SARS-CoV-2 B.1.1.7 (alpha) variant. After correction for viral load, infections with the SARS-CoV-2 B.1.1.7 (alpha) variant were 7.4 times more likely to result in a SARS-CoV-2 Ag-negative test result compared to SARS-CoV-2 infections with non-alpha lineages. This finding might have serious implications for the use of SARS-CoV-2 Ag tests, since Ag testing has been advocated as a potential aid in the solution of reopening schools, mass events, and international travel (2). The ECDC states that "the use of rapid Ag tests is appropriate in high prevalence settings when a positive result is likely to indicate true infection, as well as in low prevalence settings to rapidly identify highly infectious cases. Rapid Ag tests can help reduce further transmission through early detection of highly infectious cases, enabling a rapid start of contact tracing" (www.ecdc.europa.eu/sites/default/files/documents/Options-use-of-rapid-antigen-tests-for-COVID-19.pdf). The PanBio COVID-19 Ag test has been studied extensively, generally with favorable results. A recent Cochrane review reported an average clinical sensitivity of 75.1% in symptomatic persons (1). Previous estimates of the sensitivity of the PanBio COVID-19 Ag test in the Netherlands varied between 53 and 93% (https://lci.rivm.nl/sites/default/files/2021-02/Status%20validatie%20SARS-CoV-2%20antigeensneltesten%208%20februari%202021.pdf), depending on the population studied. Our population consisted entirely of adult HCWs with mild to moderate symptoms, who were tested in an early phase of their infection. Furthermore, the samples were taken by experienced and trained personnel of the emergency department and were processed and analyzed within one hour by experienced and trained laboratory personnel. All these factors contribute to a fairly consistent database.

A possible explanation for the lower sensitivity of the Ag test in the case of the B.1.1.7 (alpha) variant could be that mutations characterizing this variant result in a virus particle that binds less avidly to the antibodies present in the PanBio COVID-19 Ag test. As this assay targets the nucleocapsid protein, it is more likely that B.1.1.7-associated nucleocapsid mutations are responsible for the lower observed sensitivity for B.1.1.7 variants and not the B.1.1.7 characteristic N501Y mutation in the S glycoprotein. The S235F nucleocapsid mutation could be a potential candidate. Epitopes rendered from the S235F mutation proved incapable of binding antibodies generated against the original virus, which could explain false-negative test results (3). On the other hand, structure-function analyses show that the S235F occurs outside the RNA and protein binding domain, suggesting no effect on Ag test results (4). Further research is needed to elucidate the mechanism or mutations responsible for the decreased sensitivity of the B.1.1.7 variant for the PanBio COVID-19 Ag test.

The B.1.1.7 (alpha) variant has a higher infectivity because of an increased affinity

for its prime human epithelial receptor, the ACE2 receptor (5). One could also hypothesize that a lower viral burden of the B.1.1.7 (alpha) variant (compared to that of non-B.1.1.7 variants) leads to human infections, with a consequently decreased sensitivity of the Ag test. However, the multivariable logistic regression analysis suggests that both factors (the variant as well as the viral load) influence the performance of the PanBio COVID-19 Ag test, with both a higher Cp value and the B.1.1.7 (alpha) variant having a negative effect. In fact, contrary to our results, infections with the B.1.1.7 (alpha) variant (compared to infections with the non-B.1.1.7 variants) were associated with higher viral loads in a large cohort of patients in Germany (6). This difference in results could be due to earlier onset of symptoms after infection with the B.1.1.7 (alpha) variant compared to that after infection with the non-B.1.1.7 variants, in turn resulting in lower Cp values in our population and hence decreased sensitivity of the Ag test for B.1.1.7. In any case, our results suggest that both viral load and being infected by a B.1.1.7 (alpha) variant attribute to the lower sensitivity of the Ag test, the most significant contribution being the B.1.1.7 (alpha) variant.

This small study has several drawbacks. Its number of PCR-positive B.1.1.7 (alpha) variant samples is small, and therefore the exact sensitivity of the PanBio COVID-19 Ag test for this variant cannot be estimated reliably. Furthermore, the study was not set up as a prospective study with the aim of comparing the performance of the Ag test in populations infected with the non-B.1.1.7 variant and the B.1.1.7 variant. This study was a result of the observation that the performance of the PanBio COVID-19 Ag test was declining in the course of time, ultimately leading to discontinuation of this Ag test for infection control purposes in our hospital. Furthermore, the PanBio COVID-19 Ag test stipulates testing within the 7-day window of infection. Although HCWs were strongly requested to test immediately with the onset of even the mildest COVID-19-related symptoms, it cannot be excluded that, e.g., due to so-called testing fatigue, over time HCWs postponed SARS-CoV-2 testing until presentation of more severe symptoms. Lastly, the VirSNiP assay used for B.1.1.7 typing between November 2020 and January 2021 cannot distinguish between B.1.1.7 (alpha) and other N501Y-bearing strains, including B.1.351 (beta) and P.1 (gamma). In total, 7/30 B.1.1.7 (alpha) variants in this study were characterized as B.1.1.7 (alpha) based solely on the presence of the N501Y mutation. Although we cannot exclude incidental misclassification, the background prevalence of non-B.1.1.7 strains bearing the N501Y mutation in the Netherlands during that time was very low, making it unlikely to influence our conclusions. Only 27/514 (4.7%) of N501Y-bearing strains in the national SARS-CoV-2 surveillance database were non-B.1.1.7, and within our region the number was even lower (2.8%). The national SARS-CoV-2 surveillance database consists of a random sample of SARS-CoV-2 patients in the Netherlands (https://www.rivm.nl/coronavirus-covid-19/onderzoek/kiemsurveillance).

All in all, our data suggest that SARS-CoV-2 variants might develop into distinguishable viruses, each with their own dynamics regarding epidemiology, control measures, vaccination efficacy, and diagnostic characteristics. In this regard, it is worthwhile to consider the diagnostic value of Ag testing as a variable that could be dependent on the variant of the SARS-CoV-2 as well, but larger studies and studies with different Ag tests are needed to corroborate our preliminary results.

## MATERIALS AND METHODS

**Subjects and specimen collection.** BovenIJ hospital is a small community hospital in Amsterdam, the Netherlands. As part of its COVID-19 infection prevention program, health care workers (HCWs) with COVID-19-related symptoms were strongly encouraged to test for SARS-CoV-2. The COVID-19 infection control policy of the hospital was consistent throughout the entire study period. SARS-CoV-2 laboratory testing always consisted of both SARS-CoV-2 Ag testing and nucleic acid amplification testing (NAAT). This study includes all SARS-CoV-2 test results, both Ag and NAAT, obtained from HCWs during the period of 11 November 2020 to 20 April 2021. All specimens were collected by trained nurses using flocked swabs, following appropriate safety precautions. In total, 3 swabs were collected, one nasopharyngeal swab for Ag testing and one nasopharyngeal plus one oropharyngeal swab for NAAT. The two swabs for NAAT were combined into one single tube containing 3 mL universal transport medium (UTM, Becton

Dickinson, Sparks, MD, USA). Specimens for Ag testing were discarded directly after testing, and specimens for NAAT were stored at −80℃.

**SARS-CoV-2 antigen testing.** The PanBio COVID-19 Ag test (Abbott, Lake Country, IL, USA) is a membrane-based immunochromatography assay for the detection of SARS-CoV-2 nucleocapsid protein from nasopharyngeal swabs. The test was performed according to the manufacturer's instructions (reading time 15 minutes) by trained technicians of the clinical laboratory of BovenIJ hospital within one hour after specimen collection.

**SARS-CoV-2 nucleic acid amplification testing.** The Aptima SARS-CoV-2 assay for the Panther system (Hologic Inc., San Diego, CA, USA) is a transcription-mediated amplification (TMA) dual target assay that amplifies and detects two conserved regions of the SARS-CoV-2 ORF1ab gene. Amplicon detection uses chemiluminescent probes with distinct kinetic profiles: light emission with slow kinetics (glower) for both ORF1ab amplicons and light emission with rapid kinetics for the internal control (flasher). Amplification of at least one of the ORF1ab fragments generates a glower-type light signal quantified in relative light units (RLU). RLU signal strength cannot be used as a semiquantitative measure for SARS-CoV-2 RNA due to rapid saturation. According to the manufacturer, the analytical sensitivity of the Aptima SARS-CoV-2 assay is 83 copies/mL. The assay was performed by trained technicians of the clinical microbiology laboratory of OLVG Lab BV within 24 hours after specimen collection.

All SARS-CoV-2 RNA-positive samples were retested using the Realstar SARS-CoV2 RT-PCR (Altona Diagnostics GmbH, Hamburg, Germany) to obtain a semiquantitative measure as a proxy for viral load (expressed as Cp value). The Realstar SARS-CoV-2 RT-PCR is a dual target assay that amplifies and detects fragments of the S-gene (SARS-CoV-2 specific, Cy5 channel) and the E-gene ($\beta$-coronavirus lineage B specific, FAM channel). Nucleic extraction, amplification, and detection were performed with the MagNA Pure 96 system and Lightcycler 480 instrument II (Roche Diagnostics, Rotkreuz, Switzerland). According to the manufacturer, the analytical sensitivity of the Realstar SARS-CoV-2 RT-PCR is 140 copies/mL. The Realstar SARS-CoV-2 assay was performed by trained technicians of the clinical microbiology laboratory of OLVG Lab BV using stored NAAT specimens.

**SARS-CoV-2 B.1.1.7 typing.** All confirmed SARS-CoV-2 RNA-positive samples were typed to distinguish between the SARS-CoV-2 B.1.1.7 (alpha) variant versus classical SARS-CoV-2 variants not bearing the N501Y mutation. Samples obtained during the study period November 2020 to January 2021 were characterized using the melting curve-based SARS Spike 501 VirSNiP assay (TIB MOLBIOL GmbH, Berlin, Germany). All samples bearing the N501Y mutation were classified as suspected B.1.1.7 variants. For SARS-CoV-2 RNA-positive samples obtained after January 2021, the full SARS-CoV-2 genome sequences were obtained using Nanopore sequencing based on the ARTIC v3 amplicon sequencing protocol (https://artic.network). Full genome sequences were deposited at the GISAID repository (7).

**Ethical statement.** Ethical approval for this study was not required, because the study was an observational study performed on samples which were collected as part of standard hospital infection prevention policy. All subjects had given consent for inclusion in this report by an opt out e-mail procedure.

**Statistics.** As statistical software, SPSS was used (IBM Corp. Released 2020. IBM SPSS Statistics for Windows, Version 27.0. Armonk, NY: IBM Corp.). Results were analyzed by the Fisher's exact test and Student's $t$ test, with a $P$ value of $<0.05$ considered statistically significant. Multivariable logistic regression analysis was used to obtain odds ratios (OR) adjusted for Cp value.

## SUPPLEMENTAL MATERIAL

Supplemental material is available online only.

**SUPPLEMENTAL FILE 1**, XLSX file, 0.01 MB.

## ACKNOWLEDGMENTS

All authors have substantially contributed to the design of the study, the draft of the paper, and the approval of the final version. M.L.O supervised the microbiological part of the study and was responsible for the infection control policy, T.J.W.L. and W.A.R. supervised the molecular-biological part of the study, J.V. was responsible for statistical analysis, and D.E. was responsible for NGS and provided national and local data on circulating SARS-CoV-2 variants. No external funding was received.

We declare no conflicts of interest.

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
