## [Reviewer comments · Microbiology Spectrum]

Microbiology Spectrum

Comparison of the performance of the PanBio™ antigen-test in infections with two variants of SARS-CoV-2.

Marcus van Ogtrop, Thijs Van de Laar, Joost Vanhommerig, Wil Van der Reijden, and Dirk Eggink

Corresponding Author(s): Marcus van Ogtrop, OLVG Lab BV

Review Timeline:

Submission Date:	July 14, 2021
Editorial Decision:	August 18, 2021
Revision Received:	October 22, 2021
Accepted:	October 25, 2021

Editor: S. Wesley Long

Reviewer(s): Disclosure of reviewer identity is with reference to reviewer comments included in decision letter(s). The following individuals involved in review of your submission have agreed to reveal their identity: Isaac Ssewanyana (Reviewer #2)

Transaction Report:

DOI: <https://doi.org/10.1128/Spectrum.00884-21>

August 18, 2021

Dr. Marcus Lucas van Ogtrop
OLVG Lab BV
Clinical Microbiology
Oosterpark 9
Amsterdam
Netherlands

Re: Spectrum00884-21 (Comparison of the performance of the PanBio™ antigen-test in infections with two variants of SARS-CoV-2.)

Dear Dr. Marcus Lucas van Ogtrop:

Thank you for submitting your manuscript to Microbiology Spectrum. The reviewers found your manuscript has merit and is worthy of publication if a few specific issues can be addressed as detailed in their comments. When submitting the revised version of your paper, please provide (1) point-by-point responses to the issues raised by the reviewers as file type "Response to Reviewers," not in your cover letter, and (2) a PDF file that indicates the changes from the original submission (by highlighting or underlining the changes) as file type "Marked Up Manuscript - For Review Only". Please use this link to submit your revised manuscript - we strongly recommend that you submit your paper within the next 60 days or reach out to me. Detailed information on submitting your revised paper are below.

Link Not Available

Sincerely,

S. Wesley Long

Journals Department
Reviewer comments:

Reviewer #1 (Public repository details (Required)):

The study does not mention whether the full genome sequences were deposited in any public repository

Reviewer #1 (Comments for the Author):

The authors raise a pertinent question that requires looking further into.

Corrections:

1. The South African variant is B.1.351 and not B.1.135 as stated in the results.
2. The study says the patients were tested in the early phase of infection, it would be good to know how early this is i.e. detail the days from onset to collection of samples. The PanBio test kit stipulates testing within 7days of infection
3. Does the SARS Spike 501 VirSNIP assay distinguish between all variants that possess the N501Y mutation? This assay was used between November 2020 and January 2021 and it is during this time that variant B.1.351 that bears the same mutation was reported. This limitation is mentioned, and it is stated that the joint prevalence of viruses with this mutation was 1%, this

needs to be referenced

4. In the results section, it is stated that some health workers were tested more than once. How many were these and after how long was the retest done? And were the results obtained by both the Ag and PCR test on retest similar or different? This could indirectly contribute to the negative Ag tests and the positive PCR tests because the period for PCR test detection is generally longer compared to the Ag test.

5. The authors state in the discussion that the mutation S235F could explain the low sensitivity in the antigen test. This needs to be discussed objectively and with more references. One paper suggest that this mutation occurs outside the RNA and protein binding interaction interfaces and a mutation analysis done predicts a high stabilisation effect conferred by this mutation (Singh J. 2021 Structure-Function analyses of new SARS-CoV-2 variants.....). This would imply that the mutation has no effect on the Ag test. However, another paper (Ross C. 2021 Epitope specific antibody responses....) found the the S235F rendered epitopes completely incapable of binding antibodies generated against the original virus which argument would support the author's findings.

Reviewer #2 (Comments for the Author):

The author wanted to understand the cause of the seemingly reducing sensitivity of the PanBio antigen RDT that coincided with the emergence of the B.1.1.7 variant in Netherlands, by comparing the sensitivity of the antigen RDT between the B.1.1.7 and the non B.1.1.7. RealStar SERS COV-2 RT-PCR was the gold standard.

Overall the question asked in this study is important due to the unknown implication of variants on the use of Ag RDT in the rapidly evolving SARS COV-2 pandemic.

Comments for improvement

- 1.The title refers to two SARS COV-2 variants which is misleading. The author compared a variants with or without N501Y.
- 2.It is not accurate to define the B.1.1.7 variant by just one mutation. The author did not indicate how many of the 65 samples were collected in January and therefore received a full genome sequencing and if any of the non B1.1.7 and the N501Y mutation. Otherwise, it is important for the author to refer to the variants with N501Y as suspected B.1.1.7.
- 3.It is also not clear why the author excluded the B.1.135 variant from the analysis

Staff Comments:

Preparing Revision Guidelines

For complete guidelines on revision requirements, please see the Instructions to Authors at [link to page]. **Submissions of a paper that does not conform to Microbiology Spectrum guidelines will delay acceptance of your manuscript.**

Please return the manuscript within 60 days; if you cannot complete the modification within this time period, please contact me. If you do not wish to modify the manuscript and prefer to submit it to another journal, please notify me of your decision immediately so that the manuscript may be formally withdrawn from consideration by Microbiology Spectrum.

If you would like to submit an image for consideration as the Featured Image for an issue, please contact Spectrum staff.

Review comments

Title: Comparison of the performance of the PanBio™ antigen-test in infections with two Variants of SARS-CoV-2.

The author wanted to understand the cause of the seemingly reducing sensitivity of the PanBio antigen RDT that coincided with the emergence of the B.1.1.7 variant in Netherlands, by comparing the sensitivity of the antigen RDT between the B.1.1.7 and the non B.1.1.7. RealStar SERS COV-2 RT-PCR was the gold standard.

Overall the question asked in this study is critically important due to the significant implication on the use of Ag RDT in the rapidly evolving SARS COV-2 variants.

Comments for improvement

1. The title refers to two SARS COV-2 variants which is misleading. The author compared a variants with or without N501Y.
2. It is not accurate to define the B.1.1.7 variant by just one mutation. The author did not indicate how many of the 65 samples were collected in January and therefore received a full genome sequencing and if any of the non B1.1.7 and the N501Y mutation. Otherwise, it is important for the author to refer to the variants with N501Y as suspected B.1.1.7.
3. It is also not clear why the author excluded the B.1.135 variant from the analysis

Dear Editor,

Thank you for giving us the opportunity to revise our manuscript Spectrum00884-21 entitled "Comparison of the performance of the PanBio antigen-test in infections with two variants of SARS-CoV-2". We thank the reviewers for their additional and helpful suggestions. We have addressed all comments and revised the manuscript accordingly (see below). We have addressed them and revised the manuscript accordingly. We have submitted a new version of the manuscript which shows individual changes or changes sections in red.

On behalf of all authors of Spectrum00884-21
Marc van Ogtrop, MD

Reviewer #1

Comment 1.1: The study does not mention whether the full genome sequences were deposited in any public repository

Response 1.1: *Study sequences were deposited in the GISAID respiratory, but indeed this was not mentioned in the manuscript. In the revised version of the manuscript we refer to the GISAID respiratory in the materials and methods section (line 95-96), and we have provided the full list of SARS-CoV-2 strains and their corresponding GISAID accession IDs in supplementary table S1.*

Comment 1.2: The authors raise a pertinent question that requires looking further into: The South African variant (beta) is B.1.351 and not B.1.135 as stated in the results.

Response 1.2: *Unfortunately, this was a typing error that has been corrected.*

Comment 1.3: The study says the patients were tested in the early phase of infection, it would be good to know how early this is i.e. detail the days from onset to collection of samples. The PanBio test kit stipulates testing within 7 days of infection.

Response 1.3: *Health care workers were strongly requested to test directly with the onset of even the mildest COVID-19 related symptoms. The onset of symptoms generally lies within the 7 days of infection as required by the PanBio test kit. We have no indication that testing behaviour changed during the study period, but we cannot fully exclude that e.g. due to so-called testing fatigue HCWs postponed SARS-CoV-2 testing until presentation of more severe symptoms. We have added this as a possible limitation of the study in the discussion section (lines 210-213).*

Comment 1.4: Does the SARS Spike 501 VirSNiP assay distinguish between all variants that possess the N501Y mutation? This assay was used between November 2020 and January 2021 and it is during this time that variant B.1.351 that bears the same mutation was reported. This limitation is mentioned, and it is stated that the joint prevalence of viruses with this mutation was 1%, this needs to be referenced

Response 1.4: *It is certainly a drawback to use the SARS Spike 501 VirSNIP assay for the detection of the B.1.1.7 variant since it does not distinguish between different variants that bear the N501Y mutation. To address the extent of this drawback, and thus the specificity to detect the B.1.1.7 variant, we have analysed the prevalence of N501Y bearing variants (e.g B.1.351) in (i) the national SARS-CoV-2 surveillance database of the Dutch National Institute of Public Health and the environment (RIVM) and (ii) the subset of this national surveillance specifically representing our hospital. Between December 2020 – January 2021, the time period the 501 VirSNiP assay was used, 487/514 (94.7%) N501Y bearing variants in the National database were B.1.1.7 (alpha), 4.3% were B.1.351 (beta) and 0.97% were other N501Y bearing strains. In the subset from our hospital, 35/36 (97.2%) N501Y bearing variants were B.1.1.7. We have added the proportion B.1.1.7 among N501Y bearing*

strains during the time we used the VirSNiP assay to the discussion section of the manuscript, including a reference to the national SARS-CoV-2 surveillance database (lines 213-222).

Comment 1.5: In the results section, it is stated that some health workers were tested more than once. How many were these and after how long was the retest done? And were the results obtained by both the Ag and PCR test on retest similar or different? This could indirectly contribute to the negative Ag tests and the positive PCR tests because the period for PCR test detection is generally longer compared to the Ag test.

Response 1.5: *Indeed some HCWs were tested more than once. However none of the study subjects had more than one positive COVID test result. We did not include longitudinal samples of one viremic episode, and we did not encounter reinfections. We have addressed this in the manuscript (result section). Therefore multiple sampling has not influenced outcome of the comparison between the two variants of the SARS-CoV-2 (lines 116-118).*

Comment 1.6: The authors state in the discussion that the mutation S235F could explain the low sensitivity in the antigen test. This needs to be discussed objectively and with more references. One paper suggest that this mutation occurs outside the RNA and protein binding interaction interfaces and a mutation analysis done predicts a high stabilisation effect conferred by this mutation (Singh J. 2021 Structure-Function analyses of new SARS-CoV-2 variants). This would imply that the mutation has no effect on the Ag test. However, another paper (Ross C., 2021 Epitope specific antibody responses differentiate COVID-19 outcomes and variants of concern; *JCI Insight* (2021) 6:e148855) found the S235F rendered epitopes completely incapable of binding antibodies generated against the original virus which argument would support the author's findings.

Response 1.6: *The authors want to thank the reviewers for pointing out the uncertainties about the possible role of mutation S235F in false-negative COVID-19 Ag tests, and providing us with manuscripts to support and disprove our hypothesis. One study predicts a high stabilisation effect conferred by this specific SNP, whether others suggest ineffective binding of the S235F-affected epitope to (neutralizing) antibodies. We have added this to the discussion section of the manuscript (lines 179-185).*

Reviewer #2

The author wanted to understand the cause of the seemingly reducing sensitivity of the PanBio antigen RDT that coincided with the emergence of the B.1.1.7 variant in Netherlands, by comparing the sensitivity of the antigen RDT between the B.1.1.7 and the non B.1.1.7. RealStar SARS COV-2 RT-PCR was the gold standard. Overall the question asked in this study is important due to the unknown implication of variants on the use of Ag RDT in the rapidly evolving SARS COV-2 pandemic.

Comments 2.1: The title refers to two SARS COV-2 variants which is misleading. The author compared variants with or without N501Y.

Response 2.1: *We agree that for the first part of the study period (November 2020 to January 2021) we have compared SARS-CoV-2 variants with and without the N501Y mutation. However, for the second part of the study period (February 2021 – April 2021) typing was performed using full-genome sequencing and hence we were able to distinguish between B.1.1.7 and other N501Y bearing variants. In total we found 31 N501Y bearing strains, 24/31 (75%) were typed using NGS revealing 23 (95.8%) B.1.1.7 strains and 1 (4.2%) B.1.351 strain. This leaves only 7 N501Y bearing strains that were classified as 'suspected B.1.1.7' solely based on the presence of N501Y. Between November 2020 – January 2021, non-B.1.1.7 strains comprised an estimated 2.8% and 5.3% of the N501Y bearing circulating SARS-CoV-2 strains in our hospital and the Netherlands, respectively (see response 1.4). With only 7 strains classified as B.1.1.7 solely by the presence of N501Y, this suggests no or a maximum of one misclassification. Therefore, to our opinion it is acceptable to make a distinction between B.1.1.7 (alpha) versus non-B.1.1.7, We have changed the title accordingly.*

Comment 2.2: It is not accurate to define the B.1.1.7 variant by just one mutation. The author did not indicate how many of the 65 samples were collected in January and therefore received a full genome sequencing and if any of the non B1.1.7 and the N501Y mutation. Otherwise, it is important for the author to refer to the variants with N501Y as suspected B.1.1.7.

Response: *The authors have indeed not provided details on the numbers and outcome of typed strains per period (period 1: 501 VirSNiP versus period 2: NGS). In period 1 we have typed 40 SARS-CoV-2 strains using the 501 VirSNiP assay, 33 SARS-CoV-2 strains lacked the N501Y mutation and 7 strains had the N501Y mutation, These 8 strains were suspected for B.1.1.7 (alpha). In period 2, we have typed 26 SARS-CoV-2 strains: B.1.1.7 (n=23), B.1.351 (n=1), B.1.258.21 (n=1) and B.1.177.73 (n=1). Hence, in total we have 23 confirmed B.1.1.7 strains (NGS), 36 confirmed non-B.1.1.7 strains (NGS + VirSNiP) of which one B.1.351 strain that had the N501Y mutation, and 7 suspected B.1.1.7 strains (VirSNiP). As discussed in the response to comment 2.1, with the low background-prevalence of non-B.1.1.7 strains bearing the N501Y mutations it is likely that all 8 suspected B.1.1.7 strains are indeed B.1.1.7. We have clarified the results in the second paragraph of the results section (lines 128-140).*

Comment 2.3: It is also not clear why the author excluded the B.1.135 variant from the analysis

Response: *In the period December 2020 - January 2021, it was known that B.1.351 bears the N501Y mutation. It was the primary hypothesis that B.1.1.7 variant had a lower antigen sensitivity in comparison to non-VOCs. We decided to exclude other N501Y bearing VOCs (B.1.351) from the analysis (Lines 137-140). Based on the national pathogen surveillance data, it was acceptable to do this, since the frequency of other VOCs among strains with the N501Y mutation was below 5% (lines 213-222) .*

Additional revisions made by the authors - authorship

As proposed by the reviewers we included SARS-CoV-2 full genome data and we calculated the proportion of B.1.1.7 strains among SARS-CoV-2 strains bearing the N501Y mutation both in the Netherlands and in our region specifically. NGS including NGS analysis was performed by dr. D. Eggink from the National Laboratory of Public Health and the Environment in the Netherlands, and he also provided us the typing data from the national SRAS-CoV-2 surveillance database. Providing these data as well as constructive input for the manuscript warrants co-authorship, and therefore we have added dr. Eggink as the third author of our manuscript.

October 25, 2021

Dr. Marcus Lucas van Ogtrop
OLVG Lab BV
Clinical Microbiology
Oosterpark 9
Amsterdam
Netherlands

Re: Spectrum00884-21R1 (Comparison of the performance of the PanBio™ antigen-test in infections with two variants of SARS-CoV-2.)

Dear Dr. Marcus Lucas van Ogtrop:

Your manuscript has been accepted, and I am forwarding it to the ASM Journals Department for publication. You will be notified when your proofs are ready to be viewed.

Sincerely,

S. Wesley Long
Editor, Microbiology Spectrum

Journals Department
Supplemental Dataset: Accept